# A New Method for UAV Lidar Precision Testing Used for the Evaluation of an Affordable DJI ZENMUSE L1 Scanner

**Martin Štroner** **, Rudolf Urban * and Lenka Línková**

Department of Special Geodesy, Faculty of Civil Engineering, Czech Technical University in Prague, Thákurova 7, 166 29 Prague, Czech Republic; martin.stroner@fsv.cvut.cz (M.Š.); lenka.linkova@fsv.cvut.cz (L.L.)
* Correspondence: rudolf.urban@fsv.cvut.cz

**Abstract:** Lately, affordable unmanned aerial vehicle (UAV)-lidar systems have started to appear on the market, highlighting the need for methods facilitating proper verification of their accuracy. However, the dense point cloud produced by such systems makes the identification of individual points that could be used as reference points difficult. In this paper, we propose such a method utilizing accurately georeferenced targets covered with high-reflectivity foil, which can be easily extracted from the cloud; their centers can be determined and used for the calculation of the systematic shift of the lidar point cloud. Subsequently, the lidar point cloud is cleaned of such systematic shift and compared with a dense SfM point cloud, thus yielding the residual accuracy. We successfully applied this method to the evaluation of an affordable DJI ZENMUSE L1 scanner mounted on the UAV DJI Matrice 300 and found that the accuracies of this system (3.5 cm in all directions after removal of the global georeferencing error) are better than manufacturer-declared values (10/5 cm horizontal/vertical). However, evaluation of the color information revealed a relatively high (approx. 0.2 m) systematic shift.

**Keywords:** UAV; lidar; precision; accuracy



## 1. Introduction

Remote sensing methods of mass data collection are increasingly used in the current technical and scientific practice. These techniques, such as UAV or aerial photogrammetry, or 3D scanning (ground, mobile or airborne), provide point clouds, which are subsequently used for modeling of terrain or above-ground objects. For small scale applications, UAV photogrammetry combined with SfM (structure from motion) processing has been the most economical and, hence, the most widely used method the far. The difference between laser scanning and SfM photogrammetry lies mainly in the point cloud generation characteristics; while SfM reconstructs a model from visible imagery and some smoothing of surfaces can occur, laser scanning simply measures individual points on the surfaces of individual objects.

SfM is presently used in many applications including volume determination [1,2], mining [3], forestry [4], determination of solar potential [5], terrain changes [6,7], natural hazards [8,9], etc. It is, however, always necessary to consider the limitations of the methods associated with the suitable flight strategy [10–12] ground control points (GCPs) placement [13,14], processing algorithms [15], etc. Laser scanning (terrestrial, mobile or airborne) has been used in many fields including heritage conservation, [16,17], geology [18,19] river bathymetry [20], flood risk evaluation [21], archeology [22], monitoring of coastal changes [23–25], bridge constructions [26], mining areas [27], energy sector [28,29], etc. It can be reasonably expected that UAV lidar systems will, in the future, find application at least in some of these fields as well.

In view of the generally (almost prohibitively) high price of lidar scanners that can be mounted on UAVs (considering the limited payload of the drone and resulting necessity for miniaturization of the scanner and highly accurate positioning/orientation system

consisting of the inertial navigation system and GNSS RTK receiver, as well as the need for extremely accurate synchronization of all parts), they are not widely used in practice. Where used, commercial solutions such as, for example, Yellowscan Mapper [30], Phoenix Scout-16 [31], or RPLiDAR [32] are employed. These are usually more expensive but more user-friendly than "home-made" systems assembled from individual components [33–35].

Most UAV–lidar systems are based on similar laser scanners (Velodyne VLP-16, later renamed to Puck, Velodyne HDL-32, Riegl LMS-Q680i, or IBEO LUX 4L) and, therefore, have similar characteristics. An overview of laser scanners commonly used for UAV lidar can be found in [34].

Still, the total accuracy, i.e., the accuracy of determining the individual points, is the key characteristic affecting the possible utilization of any 3D system in practice. Some manufacturers provide details only on the accuracy of the individual components, i.e., partial accuracies of the laser scanner, of the GNSS RTK receiver, and the inertial measurement unit (IMU). Hence, determining the overall accuracy of UAV lidar systems is not so simple, and to obtain a meaningful result, it is necessary to compare the results with another measurement with a (known) better accuracy, such as another 3D scanner [20]. Another option is the use of control points georeferenced using GNSS RTK and/or total station [36]; nevertheless, the accuracy can be typically reliably determined only in the height component as the identification of any individual (control) point in the point cloud is unreliable due to the extremely high density of the point cloud. Torresan et al. proposed an unusual method for determining the total accuracy of a UAV–lidar system—they used the surfaces of large rectangular boxes and their intersections for determining systematic errors of point cloud coordinates (x, y, z) and standard deviations of elevation [33].

Recently, a new lidar 3D scanner DJI L1 appeared on the market at the price of Eur 12,300; together with the DJI Matrix 300 UAV and DJI Terra software, it constitutes a complete system for the acquisition of point clouds, including reflection intensities and true colors. The introduction of such (relatively) affordable lidar UAV systems will likely lead to an increase in their use, which highlights the importance of a method for independent evaluation of its accuracy that could be universally applicable to any other (new) system as well. Surprisingly, however, there are not many papers focusing on the full-scale accuracy testing of such a system and those available rarely delve deeper into the analysis of the errors.

For example, Siwiec [37] evaluated only the relative accuracy of a UAV/lidar system Ricopter/Riegl VUX-1 on flat surfaces of a bridge by intersecting a plane through the point cloud and subsequent calculation of deviations from this plane. However, such an approach can identify only one partial component of the global error associated with the real-world use of a UAV lidar system and, for this reason, we believe it to be unsuitable for deeper analysis. Salach et al. [30] investigated the elevation accuracy of UAV lidar-derived DTMs using UAV Hawk Moth with a YellowScan Surveyor scanner (based on the Velodyne VLP-16 sensor) at a 50 m flight altitude. The RMSEs ranged from 0.11 m to 0.145 m. Torresan et al. [33] evaluated the performance of a proprietary system containing a scanner LUX 4L and GNSS/INS VN-300 using intersections of planes forming the sides of rectangular boxes for the detection of reference points within the cloud. Standard deviations derived from several individual flights at a 10 m altitude ranged between 5 and 10 cm.

Hu et al. [34] tested a system consisting of a DJI Livox MID40 laser scanner, a starNeto unit (IMU), and a UAV DJI Matrice 210 carrier. Testing focused on forestry applications, the achieved accuracies were 0.5 m and worse; however, the authors did not focus on the exact evaluation of the scanning system accuracy but rather on the determination of forest stand heights). The authors also performed tests aiming to evaluate both global and relative accuracies of their system with a flight altitude of 150 m. The results were partially similar to ours—a systematic shift of approx. 3.7 cm in the horizontal and 4.7 cm in the vertical direction. Relative accuracies were determined by intersecting planes through the planar parts of objects, yielding standard deviations of 12 and 22 cm for horizontal and vertical

coordinates. From the perspective of the comparison with our results, these values are interesting; although both manufacturer-declared and global accuracies are similar to ours, these local deviations reported in their paper are much higher. Nevertheless, as the authors did not provide details of the used methods, the critical evaluation of these results and head-to-head comparison to ours is impossible.

Fuad et al. [36] evaluated the elevation accuracy of a digital terrain model using an AL3 S1000 UAV system based on a lidar sensor Velodyne HDL-32 and its dependence on the flight height and shape of the terrain. In sloped terrain, RMSEs exceeded 0.25 m even at the flight altitude of 20 m (which generally provided the best results); in flat terrain, RMSEs were as low as approx. 0.02 m regardless of the flight altitude. Considering the manufacturer-declared distance meter accuracy (0.02 m), it is likely that the high RMSEs in the sloped terrain reflect a systematic shift of the point cloud, which was not removed in their study. Pilarska et al. [38] reviewed the accuracy of many systems; however, only a theoretical analysis of errors was performed and although such an overview of available solutions is useful, the accuracy evaluation lacks practical verification.

The results of the studies described above obviate that a universal testing algorithm allowing a deeper accuracy evaluation that would also reveal the individual components of the global error is needed. In this paper, we aimed to (i) propose a method for evaluation of the accuracy of UAV lidar systems and (ii) to apply this method in practice in the evaluation of the performance of a new and relatively affordable lidar scanner DJI Zenmuse L1 mounted on UAV DJI Matrice 300.

## 2. Materials and Methods

A rugged area of a small landfill with a variety of materials including vegetation cover was selected for testing. The testing was logically divided into two basic accuracy tests: (a) the overall positioning accuracy and orientation of the point cloud in space and (b) the internal quality of the cloud.

The total positioning accuracy and orientation were determined by comparing the results with those acquired by georeferencing of individual control points using GNSS RTK and total station (expected standard deviations of GNSS RTK measurements of the horizontal and vertical coordinates in the Czech Republic are approx. 0.02 and 0.04 m, respectively).

The unambiguous identification of individual points in the L1 lidar-acquired point cloud represented the first challenge. We used a method similar to that used for validation of the first terrestrial 3D scanners (such as Cyrax 2500 or Leica HDS 3000), i.e., targets made of high reflection foil facilitating their identification in the point cloud based on the reflection intensity.

To evaluate the internal quality of the point cloud, we used a comparison with a UAV photogrammetry/SfM-acquired point cloud with significantly better accuracy (standard deviation <0.01 m, see Section 2.4.1) and point density (as will be shown in Table 5) than the lidar point cloud (the manufacturer-declared standard deviations of which for DJI L1 are 0.1 m for horizontal and 0.05 m for vertical coordinates at 50 m flight altitude).

### 2.1. Used Instruments

2.1.1. Terrestrial Measurements

Terrestrial georeferencing was performed using a robotic total station Trimble S9 (standard deviation of distance measurements is 0.8 mm + 1 ppm, that of horizontal direction and zenith is 0.3 mgon) and a GNSS RTK receiver Trimble GeoXR.

2.1.2. UAV DJI Matrice 300

The UAV DJI Matrice 300 (Figure 1a), which was used as the platform for mounting the tested laser scanner DJI Zenmuse L1 and DJI Zenmuse P1 camera, is a professional quadcopter equipped with GNSS RTK receiver. The basic characteristics of this UAV are detailed in Table 1.

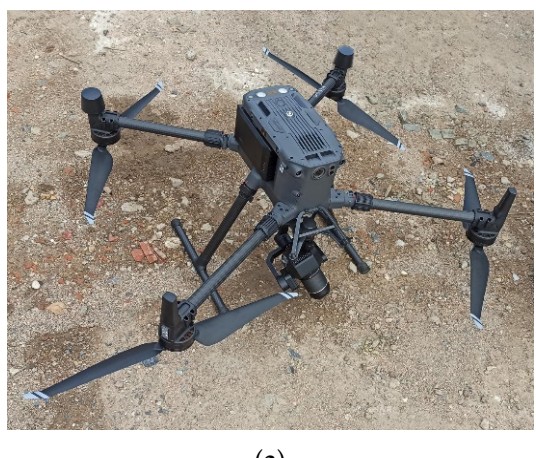

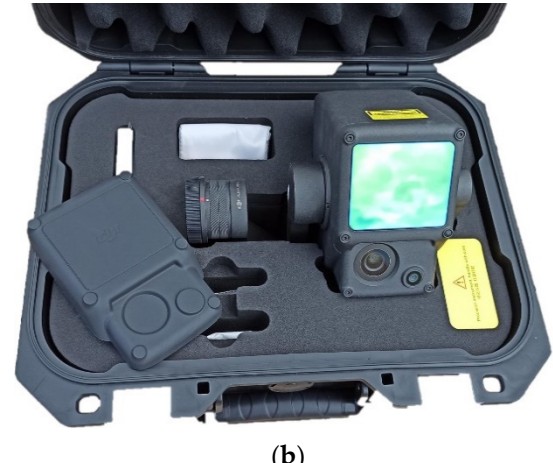

|  |  |
|:---:|:---:|
| (**a**) | (**b**) |

**Figure 1.** (**a**) DJI Matrice 300 with a DJI Zenmuse P1 camera (**b**) DJI Zenmuse L1 in the transport case.

**Table 1.** Basic characteristics of the UAV DJI Matrice 300.

| Weight | Approx. 6.3 kg (With One Gimbal) |
|:---:|:---:|
| Max. transmitting distance (Europe) | 8 km |
| Max. flight time | 55 min |
| Dimensions | 810 × 670 × 430 mm |
| Max. payload | 2.7 kg |
| Max. speed | 82 km/h |

### 2.1.3. Laser Scanner DJI Zenmuse L1

The tested laser scanner DJI Zenmuse L1 (Figure 1b) combines data from an RGB sensor and the IMU unit in a stabilized 3 axis gimbal, thus providing a true color point cloud from the RGB sensor; the point cloud must be processed in the manufacturer-supplied software DJI Terra. Basic manufacturer-declared characteristics are shown in Table 2. Since the principal aim of the paper is to propose a method for evaluation of a complex system, a detailed description and discussion of the individual parts of the DJI L1 system beyond the provided characteristics is, therefore, irrelevant. More detailed information can be found on the manufacturer's website (www.dji.com/cz/zenmuse-l1/specs, 20 November 2021).

**Table 2.** Basic characteristics of the DJI Zenmuse L1 laser scanner.

| Dimensions | 152 × 110 × 169 mm |
|:---:|:---:|
| Weight | 930 ± 10 g |
| Maximum Measurement Distance | 450 m at 80% reflectivity, 190 m at 10% reflectivity |
| Recording Speed | Single return: max. 240,000 points/s; Multiple return: max. 480,000 points/s |
| System Accuracy (1σ) | Horizontal: 10 cm per 50 m; Vertical: 5 cm per 50 m |
| Distance Measurement Accuracy (1σ) | 3 cm per 100 m |
| Beam Divergence | 0.28° (Vertical) × 0.03° (Horizontal) |
| Maximum Registered Reflections | 3 |
| RGB camera Sensor Size | 1 in |
| RGB Camera Effective Pixels | 20 Mpix (5472 × 3078) |

### 2.1.4. DJI Zenmuse P1

The DJI Zenmuse P1 camera was used for the acquisition of data for photogrammetric processing, i.e., of data used as a reference for comparison with those acquired by the DJI Zenmuse L1 scanner. This camera was developed specifically for photogrammetry; in this experiment, it was mounted with a DL 35 mm F2.8 LS ASPH lens. Basic camera characteristics are shown in Table 3, detailed information can be found on the manufacturer's website (www.dji.com/cz/zenmuse-p1/specs, 20 November 2021).

**Table 3.** Basic characteristics of the DJI Zenmuse P1 camera.

| | |
|---|---|
| Weight | 787 g |
| Dimensions | 198 × 166 × 129 mm |
| CMOS Sensor size | 35.9 × 24 mm |
| Number of Effective Pixels | 45 Mpix |
| Pixel Size | 4 μm |
| Resolution | 8192 × 5460 pix |

### 2.2. Testing Area

A rugged part of a building material landfill of approx. 100 × 100 m with materials of varying colors and ruggedness was chosen for testing. The vegetation cover was relatively minor; in the middle of the area of interest, there were two almost vertical concrete walls. Black and white photogrammetric targets and reflection targets for lidar were placed in the area, geodetic network points were stabilized by studs (Figure 2)

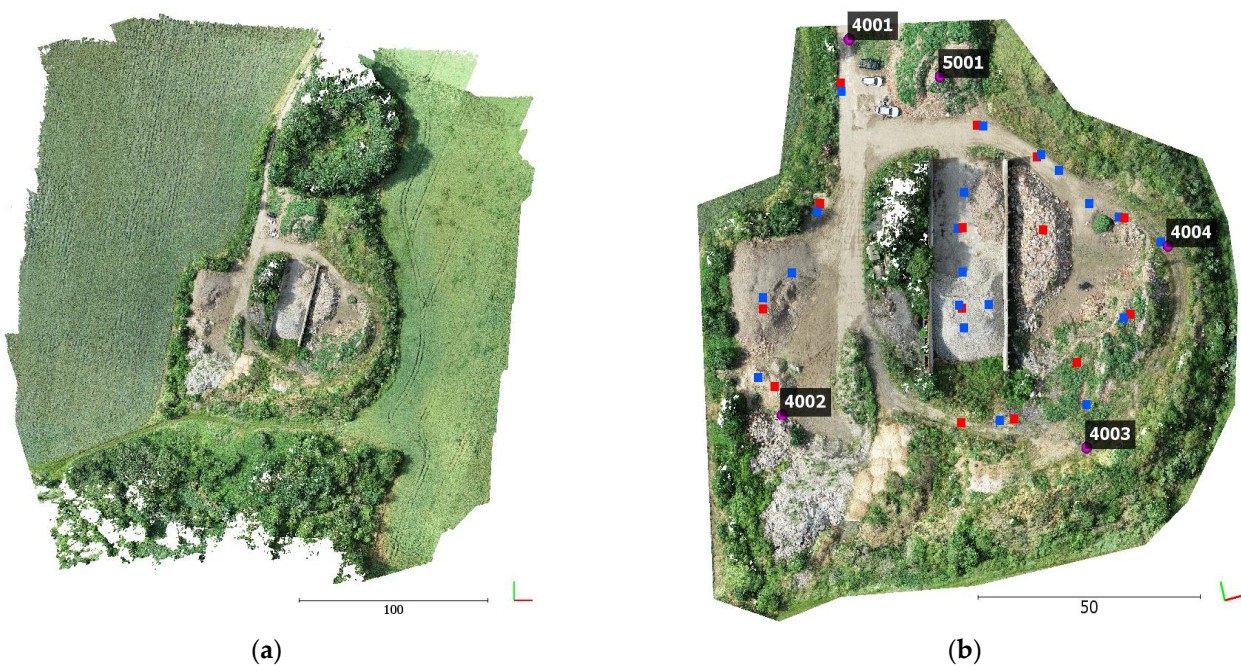

(**a**) (**b**)

**Figure 2.** (**a**) The entire scanned area; (**b**) The test area with locations of: Red—photogrammetric black and white targets; blue—reflection targets; 4001–4005—points georeferenced using the GNSS total station, 5001—position of the total station.

### 2.3. Terrestrial Measurements and Data Acquisition

2.3.1. Stabilization of (Ground) Control Points

GNSS points (see Figure 2b, points 4001–4004) were stabilized by studs. All ground control points and control points for photogrammetry were made of 40 × 40 cm hardboard and painted in a black-and-white checkerboard pattern (Figure 3a; positions marked in red in Figure 2, 14 targets in total). 20 targets for laser scanning (50 × 50 cm) were coated with high-reflection foils of one of four colors with grey or black diagonal crosses made of a 5 cm wide sticky tape (blue points in Figure 2b; see also Figure 3b–e. We tested four colors of high-reflectivity foil to determine whether they differ in reflectivity and, in effect, in their suitability for this purpose.

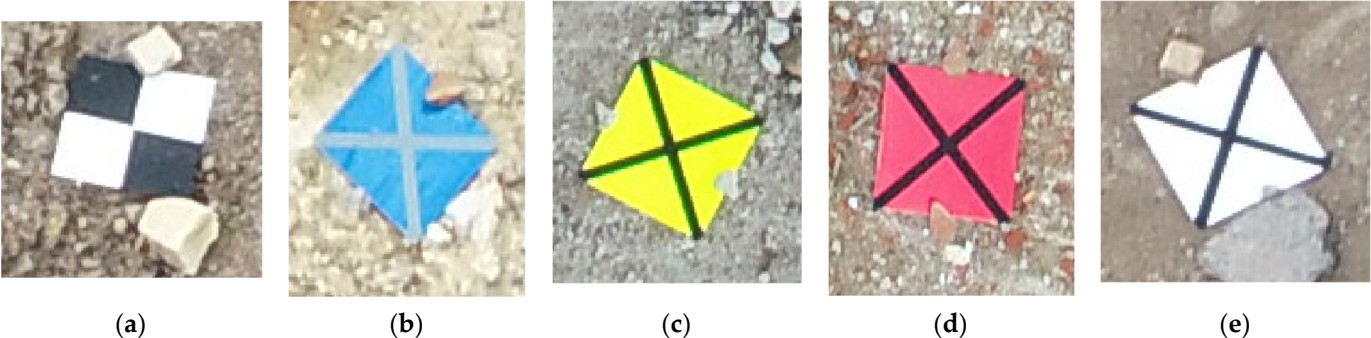

|       |       |       |       |       |
|:-----:|:-----:|:-----:|:-----:|:-----:|
| (**a**) | (**b**) | (**c**) | (**d**) | (**e**) |

**Figure 3.** Targets used as control points: (**a**) black-and-white photogrammetric target (0.4 × 0.4 m) (**b**) blue high reflection foil target (**c**) yellow high reflection foil target (**d**) red high reflection foil target (**e**) white high reflection foil target.

### 2.3.2. Terrestrial Measurements

Terrestrial measurements were performed in two faces using a total station Trimble S9 HP prior to the experiment itself (georeferencing in the local coordinate system). The positions of the centers and all corners of the hardboard target (i.e., 5 points per target) were measured for all control points and ground control points.

Four GNSS points stabilized by nails and a GNSS RTK receiver Trimble GeoXR with a 15 s observation connected to the CZEPOS permanent stations network were used for georeferencing. The accuracy of the GNSS RTK measurements was verified prior to the UAV flights, after the second flight, and after the last flight. These three measurements were used for the calculation of the standard deviation of terrestrial georeferencing (determined values of 0.0055, 0.0063, and 0.0166 m for x, y, and z coordinates, respectively).

The centers and corners of the individual targets in the JTSK coordinate system and Bpv elevation system were subsequently georeferenced based on these four GNSS points.

### 2.3.3. DJI Zenmuse L1 Data Acquisition (Lidar Data)

In all, three test flights were performed; two were at the altitude of 50 m with different data acquisition settings, the third was at the altitude of 70 m. The first flight (designated 50_1) was performed with the measurement type set to "normal" and vertical gimbal pitch of −90°; settings of the two remaining flights (designated 50 m_2 and 70 m) were oblique with gimbal pitch of −60° (i.e., 30° deviation from the downward vertical line). The remaining settings were identical: use of calibration flight—"yes"; flight speed—5 m/s; side overlap 50%; echo mode—triple; lidar sample rate—160 kHz; scan mode—repeat; rgb coloring—yes. After designating the study area, the flight planning software DJI Ground Station Pro determined the optimal path and automatically performed the flight. The individual flight paths are shown in Figure 4.

### 2.3.4. Reference Data Acquisition with DJI Zenmuse P1 (Photogrammetric Data)

Similar to the flights with lidar, the software DJI GS Pro was used for the acquisition of photogrammetric data using the DJI Zenmuse P1 camera mounted on the UAV DJI Matrice 300 RTK. The flight path (see Figure 5) was a single grid with image acquisition set to oblique (i.e., the camera on the gimbal does not maintain a constant angle but alternately acquires images in nadir and oblique directions (nadir and 15 deg in four basic directions). Thus, acquired oblique imagery greatly improves the quality of post-flight internal orientation calibration (as shown, e.g., in [10]). In all, 999 images were taken, and camera positions at the moment of the acquisition were registered using a GNSS RTK receiver connected to the CZEPOS permanent stations network.

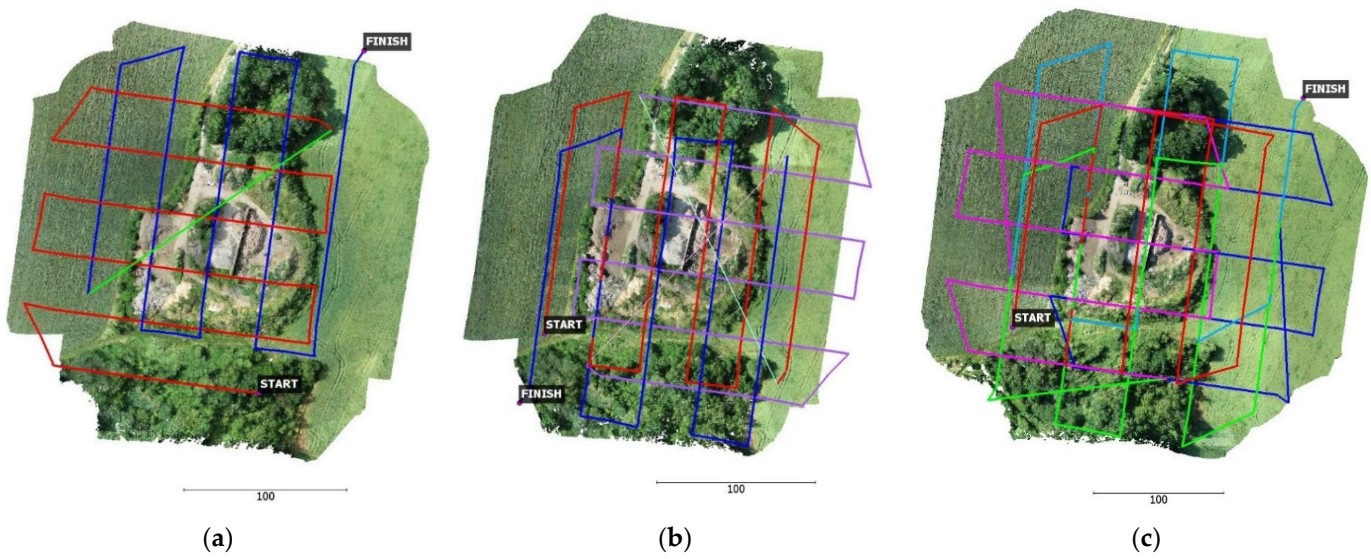

**Figure 4.** Flight paths (**a**) L1_50 m_1 flight (**b**) L1_50 m_2 flight (**c**) L1_70 m flight.

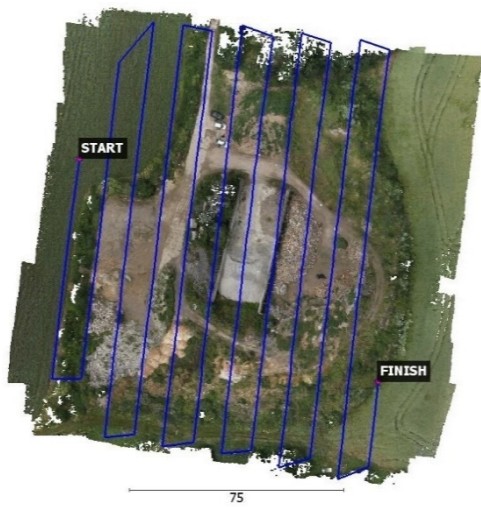

**Figure 5.** Flight path for the acquisition of the imagery for SfM processing.

### 2.4. Data Processing and Calculations

Geodetic measurements were performed in the S-JTSK coordinate system and Bpv elevation system, all other measurements were transformed using an identical algorithm into the same coordinate system (or directly calculated in the coordinate system). The coordinate system, therefore, had no influence on the result, and, in view of the size of the area of interest (only 100 × 100 m), the results are valid for any coordinate system.

#### 2.4.1. Processing of DJI Zenmuse P1 Data (Photogrammetric Data)

The reference point cloud based on DJI Zenmuse P1 imagery was georeferenced using the coordinates of five SfM targets designated as ground control points (GCPs) and camera coordinates determined by the onboard GNSS RTK receiver, considering the GCP accuracy of 0.03 m (i.e., the accuracy of GNSS RTK measurement in the Czech Republic; http://czepos.cuzk.cz, 20 November 2021). The remaining 9 SfM targets were used as check points. The calculations were performed in Agisoft Metashape ver. 1.7.1., with both Align and Dense cloud generation parameters set to "high". RMSEs of individual coordinates as well as the total error (calculated as a square root of their sum of squares) are shown in Table 4.

**Table 4.** RMSE residues after calculation—P1 flight.

|  | X [m] | Y [m] | Z [m] | Total Error [m] |
|---|---|---|---|---|
| Camera positions | 0.018 | 0.022 | 0.035 | 0.045 |
| GCPs | 0.003 | 0.002 | 0.006 | 0.007 |
| CPs | 0.002 | 0.002 | 0.011 | 0.011 |

As an additional (fully independent) method of point cloud quality evaluation, the centers of the high reflection targets from the DJI Zenmuse P1 point cloud were compared to those measured using the total station. This comparison yielded standard deviations of $s_X = 0.006$ m; $s_Y = 0.006$ m; $s_Z = 0.010$ m (total error = 0.014 m), indicating high quality. These deviations are slightly higher than accuracy characteristics determined on CPs; nevertheless, it is necessary to consider that the position was determined in a cloud with an average resolution of 12 mm (see Table 5)

**Table 5.** Numbers of points and resolutions of the point clouds.

| Flight | Number of Points (Total) | Number of Points (Cropped) | Average Point Density/m$^2$ | Resolution [mm] |
|---|---|---|---|---|
| P1 | 274,363,655 | 77,849,698 | 6551 | 12 |
| L1_50 m_1 | 85,362,025 | 21,940,068 | 1846 | 23 |
| L1_50 m_2 | 129,145,833 | 34,263,928 | 2882 | 19 |
| L1_70 m | 214,350,916 | 40,576,864 | 3413 | 17 |

These evaluations confirm a sufficiently high quality of the SfM point cloud allowing its use as a reference in view of the expected quality of the lidar point clouds. The numbers of points and other information are detailed in Table 5.

### 2.4.2. Processing of Zenmuse L1 Data (Lidar Data)

Processing of L1 data is simple—UAV data was opened in the proprietary software DJI Terra (no alternative software can be used). Next, the command New-Mission—Lidar Point Cloud Processing with preset parameters Point cloud density—High; Optimize Point Cloud Accuracy—Yes; Output Coordinate System—WGS84; Reconstruction output —PNTS, LAS triggered a calculation and exported the point cloud in the chosen format. The used LAS format recorded also additional attribute of each point, namely the RGB color, signal intensity, measurement time, order of the reflection, and other information.

Table 5 provides the basic information about the resulting point clouds.

### 2.5. Algorithms of Accuracy Assessment

As mentioned above, the accuracy assessment was divided into two parts—the global systematic positional and orientation error and the local error characterizing the remaining inaccuracies that were determined on well-controllable surfaces inside the testing area.

### 2.5.1. Global Accuracy of the Point Cloud

The global accuracy was determined using the high-intensity points (i.e., lidar targets) from the point cloud. Figure 6 shows a detail of the point cloud color-coded according to the intensities; red dots indicate high reflectivity foils.

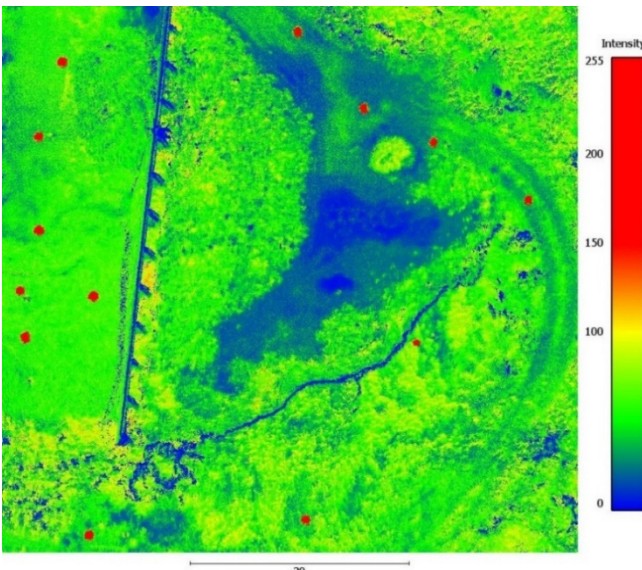

**Figure 6.** Detail of a point cloud color-coded according to the intensities; red dots indicate high reflectivity foils.

These points were manually verified; strong reflections from (e.g.,) glass shards or car bodywork were removed. The resulting point cloud was divided into partial clouds representing individual targets. For each such partial cloud, an intensity cut-off was set to yield a point cloud corresponding in size to the control point ($0.5 \times 0.5$ m). The cut-offs for yellow, white, and red targets were practically identical (intensity of 160.0) while for the blue foil, the intensities were somewhat lower, and the cut-off was set to 150.5. Subsequently, coordinates of the centers of the targets were calculated as the mean coordinates of the points with reflectivities higher than the cut-off in the individual partial point clouds/targets.

The knowledge of the coordinates of these targets based on terrestrial measurement allowed us to calculate the magnitude of systematic errors in individual coordinates, i.e., to calculate parameters of linear transformation fitting the target coordinates from the point cloud to those determined by total station measurement.

The average systematic error was calculated for each coordinate separately; in addition, we performed also the 2.5D transformation (shifts in three coordinates and a rotation about the Z-axis) and 3D transformation (3 shifts, 3 rotations). Each calculation was performed separately using the least squares method in CloudCompare 2.12 software. The transformation parameters and residual errors after transformation well described the global characteristics of the point cloud. All available points were used for each evaluation (i.e., 20 points).

### 2.5.2. Evaluation of the Local Accuracy of the Point Cloud

As the whole point cloud cannot be reliably (i.e., with the accuracy corresponding to the assumed point cloud accuracy) cleaned of vegetation points and SfM and Lidar perform highly differently in this respect, only areas free of vegetation were used for evaluation. Due to the varying character of such subsections (especially relief), they were further divided into flat (horizontal), rugged (heaps of building material, etc.) and vertical areas (walls). Each of these classes was used for the description of a different type of accuracy. The evaluation was performed using the quadratic mean of the cloud distance of individual points from the reference point cloud (i.e., SfM point cloud from DJI Zenmuse P1) and the function cloud/cloud in the CloudCompare software set to the local triangulation to 12 neighboring points. In all, 8 flat areas, 9 rugged, and two vertical areas were evaluated (see Figure 7).

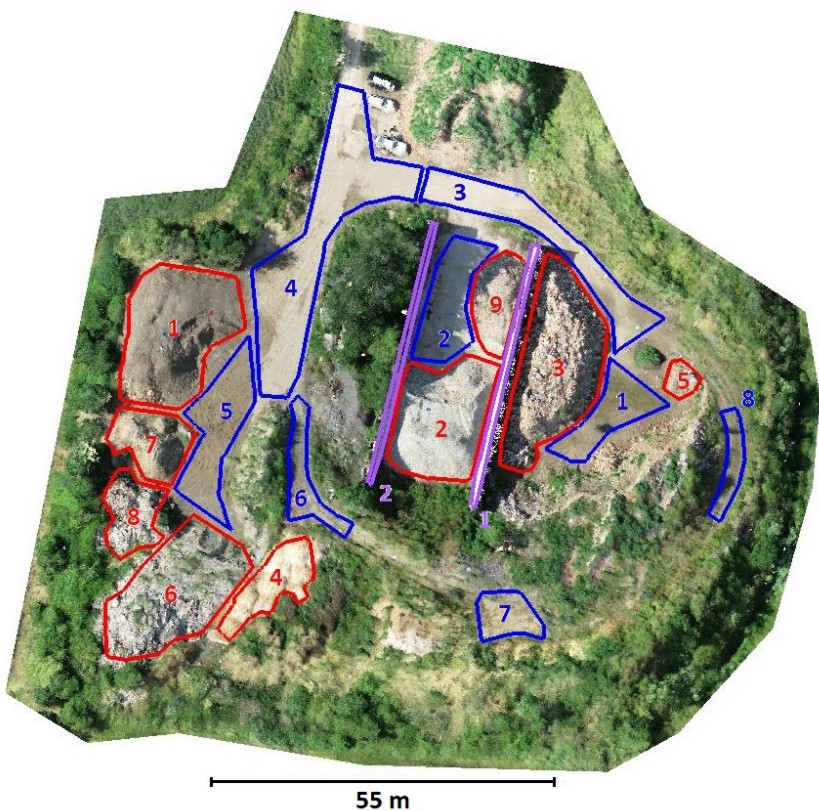

**Figure 7.** Areas of evaluation: blue—flat areas; red—rugged areas; purple—vertical areas.

### 2.5.3. Color Information Shift

During the point cloud processing, a significant shift of the color information compared to refection intensities was detected on the high reflection targets. The magnitude of the shift was calculated by determining the center of the target in the color-coded cloud as well as in the reflection intensity-based cloud and their comparison, yielding global RMSEs as well as RMSEs for individual coordinates.

## 3. Results

### 3.1. Visual Evaluation of the Point Clouds

Figure 8 shows the character of the lidar (red dots) and photogrammetric (blue) point clouds in the three types of areas. We can see that where the surfaces are relatively even, the point clouds do not differ by much (just random errors can be observed). However, where there are greater changes (higher curvatures) of the surface, we can observe additional deviations with a non- random character, i.e., systematic shift. This can be best observed in Figure 8c, where the differences are much higher near the right angle of the surface. It is possible that this fluctuation might be caused by the size of the lidar spot– the distance is determined as a mean value of the pulse and where the spot is not perfectly round (i.e., when the angle of the beam is not perpendicular to the respective surface), this mean is slightly different from the distance of the center of the beam. This would also explain the reason for the biggest error being observed on and around the sharp edges.

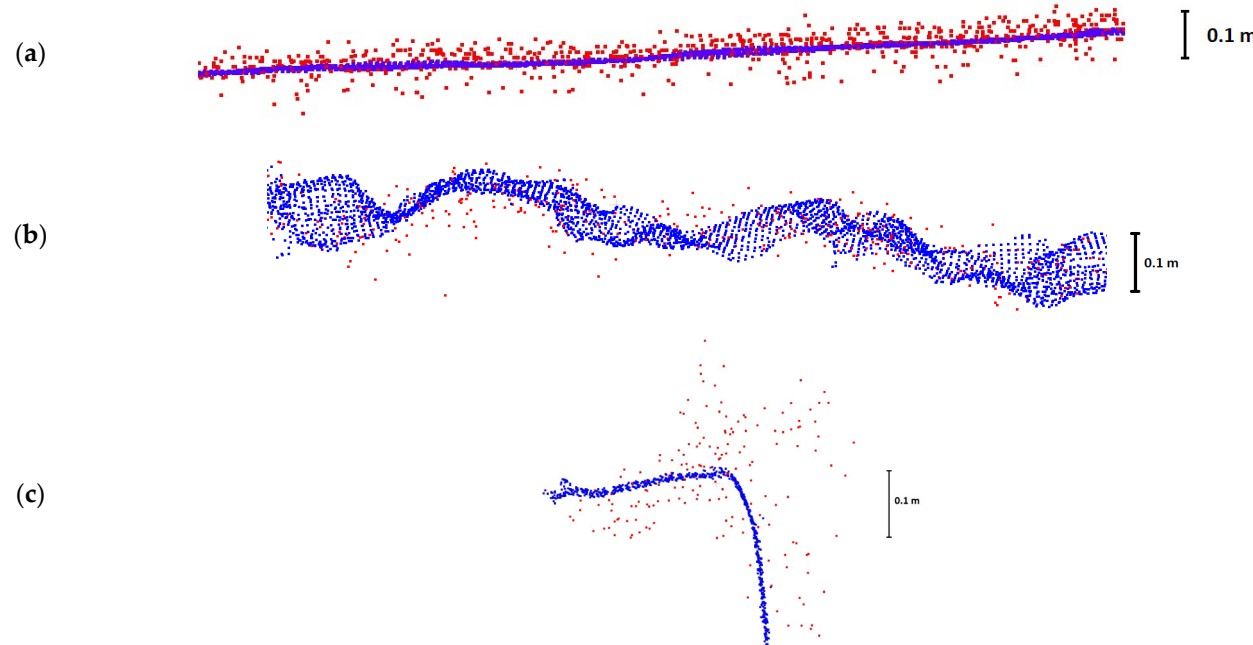

**Figure 8.** Comparison of a 20 cm wide point cloud profiles of photogrammetric data (blue) and lidar data (red), flight 50_1; (**a**) flat surface; (**b**) rugged surface; (**c**) vertical surface.

### 3.2. Global Accuracy of the Point Cloud

RMSEs of the point clouds before and after transformations are detailed in Table 6, the transformation parameters (i.e., the systematic shifts and rotation) in Table 7. RMSE is defined as:

$$RMSE_X = \sqrt{\frac{\sum (\Delta X^2)}{n}}, RMSE_Y = \sqrt{\frac{\sum (\Delta Y^2)}{n}}, RMSE_Z = \sqrt{\frac{\sum (\Delta Z^2)}{n}}, RMSE = \sqrt{\frac{RMSE_X^2 + RMSE_Y^2 + RMSE_Z^2}{3}}$$

where *X*, *Y*, *Z* are differences of checkpoint coordinates according to the reference measurement and the measurement from the particular point cloud.

**Table 6.** RMSEs compared to checkpoints without and after transformations of different types.

| Flight L1 | Type of Transformation | RMSE [m] | RMSE$_X$ [m] | RMSE$_Y$ [m] | RMSE$_Z$ [m] |
|---|---|---|---|---|---|
| 50 m_1 | Original cloud | 0.036 | 0.054 | 0.019 | 0.022 |
| | Translation | 0.013 | 0.016 | 0.013 | 0.007 |
| | 2.5D transformation | 0.013 | 0.016 | 0.013 | 0.007 |
| | 3D transformation | 0.012 | 0.016 | 0.013 | 0.005 |
| 50 m_2 | Original cloud | 0.025 | 0.024 | 0.020 | 0.030 |
| | Translation | 0.015 | 0.016 | 0.017 | 0.012 |
| | 2.5D transformation | 0.015 | 0.016 | 0.016 | 0.012 |
| | 3D transformation | 0.014 | 0.016 | 0.016 | 0.010 |
| 70 m | Original cloud | 0.029 | 0.035 | 0.030 | 0.019 |
| | Translation | 0.014 | 0.019 | 0.010 | 0.012 |
| | 2.5D transformation | 0.014 | 0.019 | 0.011 | 0.012 |
| | 3D transformation | 0.013 | 0.019 | 0.011 | 0.007 |

**Table 7.** Transformation parameters.

| Flight L1 | Type of Transformation | $T_X$ [m] | $T_Y$ [m] | $T_Z$ [m] | $R_z$ [°] | $R_x$ [°] | $R_y$ [°] |
|---|---|---|---|---|---|---|---|
| 50 m_1 | Original cloud | | | | | | |
| | Translation | 0.052 | −0.014 | 0.021 | | | |
| | 2.5D transformation | 0.052 | −0.014 | 0.021 | −0.0001 | | |
| | 3D transformation | 0.052 | −0.014 | 0.021 | −0.0001 | −0.0084 | −0.0080 |
| 50 m_2 | Original cloud | | | | | | |
| | Translation | 0.018 | −0.011 | 0.027 | | | |
| | 2.5D transformation | 0.018 | −0.011 | 0.027 | 0.0046 | | |
| | 3D transformation | 0.018 | −0.011 | 0.027 | 0.0046 | −0.0135 | −0.0053 |
| 70 m | Original cloud | | | | | | |
| | Translation | 0.030 | −0.028 | 0.015 | | | |
| | 2.5D transformation | 0.030 | −0.028 | 0.015 | −0.0050 | | |
| | 3D transformation | 0.030 | −0.028 | 0.015 | −0.0050 | −0.0126 | −0.0212 |

The difference between the global position of the original point cloud and terrestrial georeferencing approximately corresponds to the accuracy of the GNSS RTK measurements (i.e., approx. 1 to 3.5 cm), with the maximum value of more than 5 cm reported for a single coordinate in Flight 1. The mean RMSEs for individual coordinates are 3.6, 2.5 and 2.9 cm (for flights 50 m_1, 50 m_2, and 70 m, respectively). After the application of the determined systematic shift values on the data (i.e., point cloud transformation to clean the data of the systematic shift), we observe a significant improvement of all RMSEs for individual coordinates as well as of the global RMSEs for individual flights to 1.5 cm or less. The total systematic error of point cloud georeferencing differed between flights; however, after transformation using determined systematic shifts for individual flights, the agreement of the lidar point clouds with the reference geodetic measurements dropped to vary similar values of approx. 1.5 cm in all flights.

The use of additional transformations (i.e., in our case, using the angle of rotation about the vertical axis and angles of rotation about the remaining two axes) practically did not improve the results further (RMSEs improved by 1 mm or less); this implies that our experiment was not burdened with any significant error in the orientation (rotation) of the entire point cloud. The angles detailed in Table 7 may appear relatively large; nevertheless, considering the flight altitude, we still speak of only a few centimeters (max. 2.5 cm).

### 3.3. Local Accuracy on the Vegetation-Free Areas

After cleaning the point clouds of the global shift as described above, the point clouds were compared against the SfM cloud; the overall results are shown in Table 8 (detailed results in Appendix A).

**Table 8.** RMSE of lidar point clouds vs. SfM point cloud in individual types of areas.

| Flight L1 | Flat Surfaces | Rugged Surfaces | Vertical Surfaces |
|---|---|---|---|
| 50 m_1 | 0.032 | 0.038 | 0.038 |
| 50 m_2 | 0.030 | 0.038 | 0.027 |
| 70 m | 0.044 | 0.048 | 0.049 |

RMSEs on the flat surfaces show minor differences between individual areas; however, at the flight altitude of 50 m, these are close to the accuracy of the distance meter itself (manufacturer-declared accuracy of approx. 3 cm). At the flight altitude of 70 m, this error is about half again this number (4.4 cm).

The situation is similar where rugged surfaces are concerned; the errors are, however, somewhat higher (an increase from approx. 3 to 3.8 cm and from 4.4 to 4.8 cm at the 50 m and 70 m flight altitudes, respectively.

However, in the case of vertical surfaces, we observed a difference between the nadir (flight 50 m_1) and oblique (50 m_2) data acquisition. The higher error observed for the 50 m_1 flight indicates that for vertical surfaces, the vertical (nadir) acquisition is likely to provide poorer results than the oblique one.

### 3.4. The Shift of the Color Information

The color information shifts against signal intensity are shown in Table 9. The shift was relatively high, with the total error exceeding 2 decimeters. The mean shift indicates that the color information is systematically biased and, in addition, changes within the point cloud (the latter is characterized by a mean standard deviation of approx. 4.9 cm).

**Table 9.** Color information shifts for individual flights.

| Flight L1 | Shift | X (m) | Y (m) | Z (m) | Distance (m) |
|---|---|---|---|---|---|
| 50 m_1 | Mean | 0.076 | −0.196 | 0.000 | 0.210 |
|  | St. dev | 0.028 | 0.036 | 0.042 |  |
| 50 m_2 | Mean | −0.271 | −0.163 | −0.102 | 0.332 |
|  | St. dev | 0.069 | 0.051 | 0.027 |  |
| 70 m | Mean | −0.221 | 0.335 | −0.139 | 0.425 |
|  | St. dev | 0.054 | 0.071 | 0.043 |  |

Figure 9 illustrates the color information shift on the example of the flight 50 m_1. Figure 10 shows an overlay of the color information (blue reflective foil) and point cloud colored by intensities.

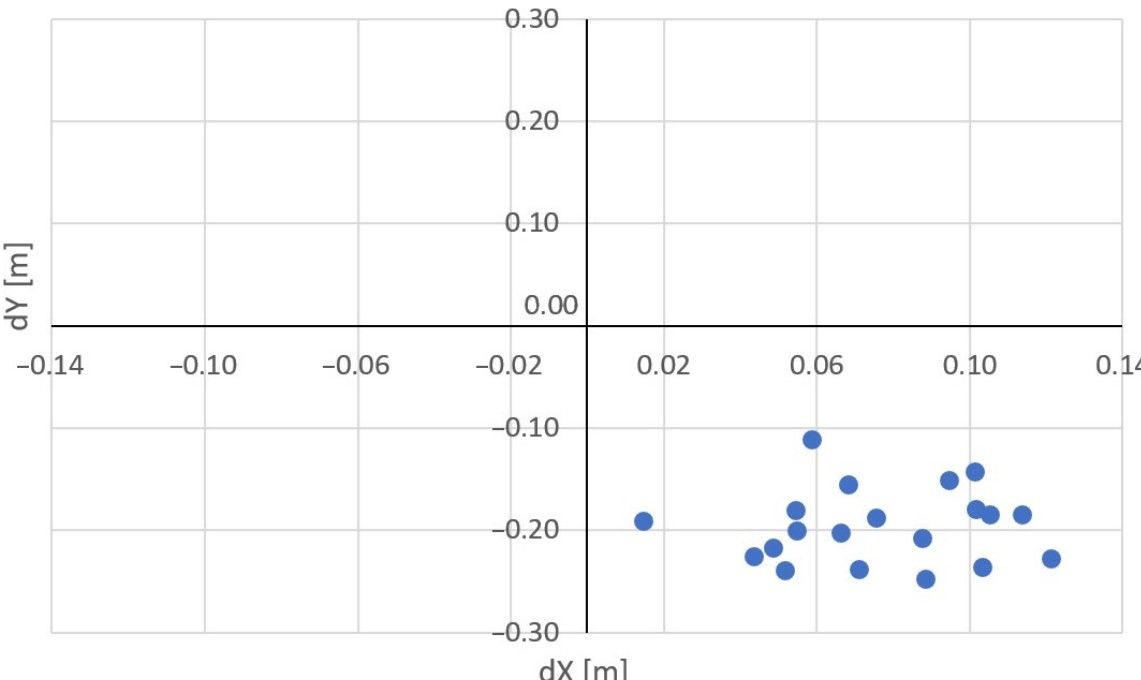

**Figure 9.** Color information shifts for individual targets acquired during the 50 m_1 flight; the placement of all shifts in a single quadrant highlights the systematic character of the error.

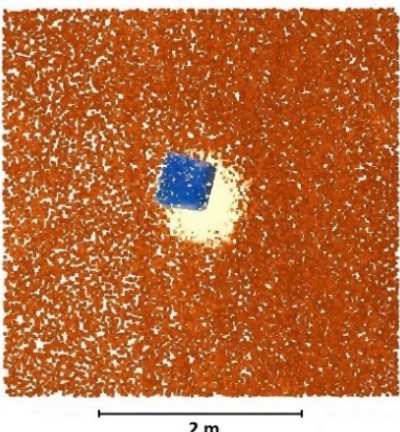

**Figure 10.** An example of the color information shift (blue target) compared to the reflection intensity (brown/yellow scale); the target centers differ by 0.25 m.

## 4. Discussion

The presented paper proposed an algorithm for a full evaluation of a UAV lidar system overcoming the problem of identification of individual points within the cloud. The principal step lies in the use of portable square targets ($0.5 \times 0.5$ m) covered with a highly reflective foil facilitating their easy identification in the point cloud as well as a relatively simple determination of the center of the target. This subsequently allows researchers to calculate the georeferencing error and remove it from the cloud through its spatial transformation. Here, the accuracy testing was performed on vegetation-free surfaces to prevent the effects of additional factors (e.g., vegetation removal). Besides smooth horizontal surfaces, we used also vertical surfaces and rugged areas (with surfaces both slopy and rough). Using this approach, we performed full testing of the DJI Zenmuse L1 system carried by a UAV DJI Matrice 300. Unlike many studies mentioned above, our results correspond to those provided by the manufacturers, especially where the lidar sensor accuracy is concerned—the RMSEs detected in our study were approx. 38 mm at the flight altitude of 50 m and approx. 48 mm at the flight altitude of 70 m, respectively (after removal of the georeferencing global error using transformation of the entire point cloud).

In our case, the determination of a simple systematic shift in all axes and its removal from the cloud was sufficient for the acquisition of optimal results; the highest observed systematic shift was 50 mm in a single coordinate; typical shifts ranged between 20 and 30 mm, which corresponds to the GNSS RTK measurement accuracy of the UAV onboard receiver. Had the global georeferencing error not been removed, RMSE would be at least twice as high (and even higher in sloped and rugged objects).

Our testing also showed that the global georeferencing error is not negligible (in our case, the georeferencing error was of approx. the same size as that of the lidar scanner). The use of the presented targets can help in the detection and subsequent significant reduction of this component of the error and can be applied both when evaluating a UAV lidar system and when performing routine measurements.

We evaluated our testing methods using three data acquisition flights. The basic flight altitude was set to 50 m as the manufacturer declared characteristics were valid for this altitude. As the manufacturer does not detail the mode of image acquisition (nadir vs. oblique), we performed both of these. In addition, to be able to evaluate the effect of the flight altitude on the accuracy, we also opted for a higher flight altitude of 70 m. Our results show that the mode of acquisition has practically no effect on the system performance while the higher altitude was associated with a slight worsening of the accuracy. This can be expected as the measured distances are longer and the lidar spot is larger, thus registering means from a greater area.

The fact that the color information provided by the lidar system is systematically positionally shifted is important for potential practical applications utilizing this information, such as the detection of ground points under the canopy based on differences in color. This shift could possibly be caused by the fact that registration of the color information (i.e., photo acquisition) is performed in certain intervals and the UAV system travels and records points even between any two acquisitions of color information. We cannot generalize this result of our study as it is valid only for the tested system; nevertheless, it seems logical that the same would apply to other systems as well. The manufacturer of our system does not provide information about the color information accuracy in the specifications, providing only the general information about the scanner accuracy, which is, in our case, better than that of color information; this might be misguiding and lead users to believe that the color information accuracy is the same.

## 5. Conclusions

We proposed a method for evaluating the quality of point clouds acquired using a lidar/UAV system. This method solves the problem of the identification of reference points within a dense point cloud using special targets with high reflectivity. Here, we summarize the key points of our approach:

1. Testing should be carried out in an area with surfaces that are approximately horizontal, vertical, and generally rugged, all without vegetation.
2. Prior to data acquisition, targets covered with a reflective film should be placed in the area and their coordinates determined using a reference method with accuracy superior to the expected accuracy of the tested system.
3. The UAV–lidar system point cloud should be supplemented with a reference point cloud (e.g., SfM as in our case) of the test area with significantly higher accuracy and detail.
4. The coordinates of the centers of targets are determined in the cloud using reflection intensities. Using these data, the systematic georeferencing error is calculated and removed from the cloud by linear transformation.
5. The resulting point cloud accuracy is determined as the RMSE of the distances between the reference (in our case, SfM) and tested (lidar) clouds for individual surfaces.

In addition to accuracy testing, the targets mentioned above can be used also to check or improve georeferencing of the entire cloud during routine measurements. Admittedly, this requires an additional processing step; on the other hand, it can further improve the accuracy of such routine measurements. Hence, the researchers should consider the level of accuracy they need in every individual case and if high accuracy is needed, this method should be used.

Results of our testing confirm that the achieved standard deviation (cleaned of the georeferencing error) is better than that declared by the manufacturer. Conversely, they have also shown the presence of this systematic georeferencing error, which is not negligible. The proposed method can help eliminate this component of error from the cloud (or, at least, significantly reduce it).

Automatic color coding of the point cloud by the tested lidar–UAV system is burdened with a relatively high (approx. 20 cm) positional inaccuracy, which may pose a problem for applications utilizing this information.

**Author Contributions:** Conceptualization, M.Š. and R.U.; methodology, M.Š. and R.U.; formal analysis, L.L.; investigation, L.L. and R.U.; data curation, M.Š.; writing—original draft preparation, M.Š., R.U., and L.L.; writing—review and editing, L.L. All authors have read and agreed to the published version of the manuscript.

**Funding:** This research was funded by the Grant Agency of CTU in Prague—grant number SGS21/053/OHK1/1T/11 "Optimization of acquisition and processing of 3D data for purpose of engineering surveying, geodesy in underground spaces and 3D scanning".

**Conflicts of Interest:** The authors declare no conflict of interest. The funders had no role in the design of the study; in the collection, analyses, or interpretation of data; in the writing of the manuscript, or in the decision to publish the results.

## Appendix A. Detailed Results—RMSE of Lidar Point Clouds vs. SfM Point Cloud

**Table A1.** RMSE of lidar point clouds vs. SfM point cloud on "flat" surfaces.

| Flight L1 | 1 | 2 | 3 | 4 | 5 | 6 | 7 | 8 | All |
|---|---|---|---|---|---|---|---|---|---|
| 50 m_1 | 0.033 | 0.032 | 0.030 | 0.031 | 0.035 | 0.036 | 0.025 | 0.028 | 0.032 |
| 50 m_2 | 0.031 | 0.028 | 0.029 | 0.027 | 0.033 | 0.034 | 0.028 | 0.029 | 0.030 |
| 70 m | 0.052 | 0.037 | 0.039 | 0.044 | 0.048 | 0.045 | 0.041 | 0.045 | 0.044 |

**Table A2.** RMSE of lidar point clouds vs. SfM point cloud on "rugged" surfaces.

| Flight L1 | 1 | 2 | 3 | 4 | 5 | 6 | 7 | 8 | 9 | All |
|---|---|---|---|---|---|---|---|---|---|---|
| 50 m_1 | 0.036 | 0.028 | 0.036 | 0.027 | 0.038 | 0.039 | 0.038 | 0.068 | 0.031 | 0.038 |
| 50 m_2 | 0.037 | 0.029 | 0.035 | 0.028 | 0.034 | 0.044 | 0.038 | 0.051 | 0.033 | 0.038 |
| 70 | 0.052 | 0.037 | 0.045 | 0.036 | 0.049 | 0.052 | 0.051 | 0.064 | 0.039 | 0.048 |

**Table A3.** RMSE of lidar point clouds vs. SfM point cloud on "vertical" surfaces.

| Flight L1 | 1 | 2 | All |
|---|---|---|---|
| 50 m_1 | 0.025 | 0.052 | 0.038 |
| 50 m_2 | 0.029 | 0.027 | 0.027 |
| 70 | 0.045 | 0.053 | 0.049 |

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
