# Peer review of "A New Method for UAV Lidar Precision Testing Used for the Evaluation of an Affordable DJI ZENMUSE L1 Scanner"

_remotesensing, doi:10.3390/rs13234811_

Round 1

Reviewer 1 Report

My overall feeling about this paper is that it is more like a technical review rather than an academic publication. The authors used lots of black boxes, e.g. SfM, lidar, without proper definition and explanations. A lot of appropriate citations are missing. Some comments:

  1. Extensive editing of English language and style required. The paper, especially the discussion/conclusion section, is hard to follow. Some grammatical and typo mistakes are in the abstract, for example, Line 12, I(n) this paper.
  2. Please include proper references to support the statement of “a UAV photogrammetry / SfM acquired point cloud with significantly better accuracy and point density (standard deviation < 0.01m)”.
  3. Why is the flight altitude 50m and 70m? Why not 10 /20 or even 100? And why 50m was flown twice, and 70m only did once?
  4. Need a section to discuss the technologies inside DJI lidar. For example, is it a rotational lidar or solid-state lidar? Is it time-of-flight based or phase-based? Any software / AI algthioms behind to optimise the results? Did you test and evaluate Velodyne, Riegl, IBEO? If yes, any results? If no, why not? The lidar used in the paper is photogrammetry optimised. This key information is missing.
  5. Please include references to support the statement of “the GCP accuracy of 0.03 m” in Line 208 / 209.
  6. The comparison of “Vertical area (walls)” is not appropriate. Structure from Motion and Lidar is to different technologies. Authors need to understand the principles of SfM and the technologies behind lidar.
  7. The first four paragraphs of Section 4 should be in the literature review, not in the discussion.

Author Response

Dear Reviewer 1

The response to the review is in the attached pdf.

Reviewer 2 Report

In this paper the accuracy of a particular aerial lidar acquisition model was investigated and compared with the photogrammetry counterpart. 

There are some useful advices to readers that want to test the equipment. Please make the RMSE errors defined accurately so as the reader to understand what you are comparing with. Maybe a mathematical formula can be handy.

Also the images with red and blue comparison of points need to be explained more why do we see this flunctuations in the lidar data? There should be pointers with bullets explaining the differences.

Apart from that everything is clear. 

Author Response

Dear Reviewer 2

The response to the review is in the attached pdf.

Round 2

Reviewer 1 Report

The explanation of "Need a section to discuss the technologies inside DJI lidar....." is ok. But you need to include this in the paper, tell the reader this has been not done yet. And this is the limitation / future work.

Structure from Motion requires the object in the camera field of view. The image needs a good quality to let SfM extract unique features to create points cloud. Given the flight altitude, the drone is not be able to capture enough features from two sides of the vertical walls. So the point cloud quality from SfM might even worse than lidar. Therefore, you can't use the SfM results of the vertical wall as the ground truth.

Author Response

Dear reviewer 1,

The response to your review is in the attached pdf.
